# Autophagy Inhibition via Hydroxychloroquine or 3-Methyladenine Enhances Chemotherapy-Induced Apoptosis in Neuro-Blastoma and Glioblastoma

**DOI:** 10.3390/ijms241512052

**Published:** 2023-07-27

**Authors:** Darcy Wear, Eesha Bhagirath, Arpana Balachandar, Caleb Vegh, Siyaram Pandey

**Affiliations:** 1Department of Chemistry and Biochemistry, University of Windsor, Windsor, ON N9B 3P4, Canada; wear@uwindsor.ca (D.W.); ebhagira@uwo.ca (E.B.); balach11@uwindsor.ca (A.B.); veghc@uwindsor.ca (C.V.); 2Department of Pharmacology and Toxicology, University of Toronto, Toronto, ON M5R 0A3, Canada; 3Brain Health Imaging Centre, Centre for Addiction and Mental Health, Toronto, ON M5T 1R8, Canada; 4Public Health, Schulich School of Medicine and Dentistry, University of Western Ontario, London, ON N6A 3K7, Canada; 5Department of Medicine, University of Toronto, Toronto, ON M5R 0A3, Canada

**Keywords:** autophagy, cancer, chemotherapy, homeostasis, hydroxychloroquine, mitochondria, oxidative stress, rapamycin, therapeutics

## Abstract

Neuroblastoma is the most common tumour in children under 1 year old, accounting for 12–15% of childhood cancer deaths. Although current treatments are relatively efficacious against this cancer, associated adverse effects could be detrimental to growth and development. In contrast, glioblastoma accounts for 52% of brain tumours and has an extremely poor prognosis. Current chemotherapeutics include temozolomide, which has numerous negative side-effects and a low-effective rate. Previous studies have shown the manipulation of autophagy to be a promising method for targeting cancers, including glioblastoma. We sought to determine the effects of autophagic alterations in combination with current chemotherapies in both neuroblastoma and glioblastoma. Supplementing cisplatin or temozolomide with autophagy activator rapamycin stabilized cancer cell mitochondria, despite having little effect on apoptosis or oxidative stress. Autophagy inhibition via 3-methyladenine or hydroxychloroquine alongside standard chemotherapies enhanced apoptosis and oxidative stress, with 3-methyladenine also disrupting mitochondrial health. Importantly, combining hydroxychloroquine with 0.5 µM cisplatin or 50 µg/mL temozolomide was as or more effective than 2 µM cisplatin or 100 µg/mL temozolomide alone. Analyzing these interesting results, a combined treatment of autophagy inhibitor with a standard chemotherapeutic agent could help to improve patient prognosis and reduce chemotherapy doses and their associated side-effects.

## 1. Introduction

Neuroblastoma (NB) is one of the most common embryonic tumors that begins from immature nerve cells, making it a tumor of the peripheral sympathetic nervous system. It is an extracranial solid tumor that exhibits unique features like early onset, representing around 8–10% of all childhood cancers, and a high tendency for spontaneous regression [1]. Commonly affecting children 5 years old and younger, it is one of the most prevalent pediatric cancers characterized by a poor prognosis resulting in 15% of all pediatric cancer fatalities [2]. In the past 30 years, the 5-year survival rate for NB patients has improved from 52% to 74%; however, approximately 55% of patients in the high-risk group experience relapse [3].

In contrast, glioblastoma (GBM) is one of the most common and aggressive malignant brain tumors diagnosed in adults, occurring in the spinal cord or brain. This cancer makes up about 16% of all primary brain and central nervous system tumors and 45.2% of malignant primary brain tumors [4,5]. Current treatments for GBM are highly ineffective resulting in a median survival of only 15 months [6]. The initial treatment is typically surgery, as this is the best and easiest way to remove the tumor barring prior metastasis. However, if the tumor has metastasized, multiple surgeries or a combination of other treatment methods may be required, including chemotherapies like cisplatin and temozolomide (TMZ), used for NB and GBM respectively, and radiation therapy [7,8,9]. Although radiation and chemotherapy drugs are an effective way to kill cancer and control further growth, they also cause intensive damage to surrounding healthy cells and tissues resulting in many unfavorable side-effects to the patient. According to the Canadian Cancer Society, these include low blood cell count, hearing problems, infections, alopecia, thinking and memory changes, inflamed mucous membranes, anemia, and an increased risk of developing a second cancer [10]. Though most side-effects are temporary, hearing issues, thinking and memory changes, inflamed membranes, and increased risk of secondary cancers can last for long periods of time or become permanent. This is of particular importance in young children battling with NB as severe damage could hinder their growth and development. The poor prognosis and significant side-effects associated with these cancers and chemotherapeutics highlight the vital need to improve current treatments. One potential method to improving patient outcome is through the manipulation of autophagy in conjunction with standard chemotherapeutic treatments.

Autophagy is a pro-survival, self-degradative process that plays a housekeeping role in the cell as it removes aggregated and misfolded proteins or dysfunctional cell components that could negatively impact cellular homeostasis [11]. Autophagy initiation occurs when the cell recognizes an aberrant protein or organelle that requires removal and results in the formation of a double membraned vesicle called a phagophore [12]. Next, nucleation involves the localization of autophagic proteins to the phagophore promoting the continuation of autophagy. Elongation results in phagophore growth eventually forming an autophagosome around the entire protein/organelle. The autophagosome fuses with an acidic lysosome forming an autophagolysosome where the lysosome degrades the autophagosome contents. Another critical cellular process, apoptosis (programmed cell death) is crucial for overall health as it ensures proper functioning and causes minimal damage to surrounding tissues [13]. Both autophagy and apoptosis are vital for optimal cellular functioning during encounters with internal or external stressors including uncontrolled cell growth, high levels of oxidative stress, and dysfunctional organelles.

A variety of autophagic modulators including rapamycin, 3-methyladenine (3-MA), and hydroxychloroquine (HCQ) may improve anti-cancer activity in combination with current chemotherapies in NB (SH-SY5Y) and GBM (U-87 Mg) cells. Rapamycin, an autophagy activator, works through the inhibition of mammalian target of rapamycin (mTOR) which results in the formation of a phagophore and subsequent autophagosome [14]. In contrast, 3-MA and HCQ are early and late-stage autophagy inhibitors respectively. Specifically, 3-MA inhibits class III phosphatidylinositol 3-kinase which mediates vesicle nucleation [15,16] whereas HCQ prevents vesicular fusion between the autophagosome and lysosome resulting in the build-up of autophagic vacuoles [17].

Due to the crucial nature of autophagy as a cell maintenance mechanism, alteration of this pathway can lead to a plethora of problems in the cell. In relation to cancer, defective autophagic machinery is a major causative effect involved in initial cancer tumorigenesis [18]. Conversely, autophagy in established tumors helps cancer cells withstand metabolic stressors and resist death induced by chemotherapy. Moreover, it allows cells to maintain viability in periods of stress which can lead to tumor dormancy, progression, and therapeutic resistance [19]. 

Although much work has been done on autophagy modulation in NB and GBM, the optimal strategies in which autophagy modulation combined with chemotherapy can be used as an anti-cancer treatment is not understood. Common chemotherapies cisplatin and TMZ have been shown to induce both dose-dependent apoptosis and autophagy in NB and GBM cells respectively [20,21]. Various groups have shown that supplementation of HCQ in NB or GBM cells sensitizes them to chemotherapy and increases apoptotic levels [20,22]. Another study demonstrated that 3-MA induced cell injury and decreased cell viability further supporting the idea that autophagy can act as a cytoprotective mechanism, and its inhibition may promote apoptosis in NB [23]. Alternatively, rapamycin had no significant effect on cell viability but did provide neuroprotection to NB SH-SY5Y cells against amyloid-β-induced toxicity, oxidative stress, and neurotoxicity. However, other studies have shown rapamycin’s potential as a possible therapeutic agent to treat NB and GBM as it halts cell proliferation and induces cytotoxicity [24,25].

Since many anti-cancer drugs such as chemotherapeutic agents create cytotoxic stress in the cell, they activate autophagy which, if performed for an excessive period, could result in cancer cell death [18]. Conversely, it has been found that the inhibition of autophagy enhances anti-cancer drug induced cell death, suggesting a novel therapeutic method to treat cancer. Clearly, a knowledge gap exists about the role of autophagy in tumor survival or suppression in these different cancers. This paper aims to investigate whether autophagy modulation can be an effective treatment in combination with current chemotherapies along with further mechanistic elucidation. We hypothesize that the inhibition of autophagy alongside standard chemotherapy regimens will result in enhanced apoptotic levels and could be utilized to reduce chemotherapy doses in NB and adverse effects in children, as well as increasing the lifespan/quality of life for GBM patients.

## 2. Results

### 2.1. Cisplatin-Induced Autophagy Is Reduced in SH-SY5Y Neuroblastoma Cells by 3-MA Supplementation

Autophagy induction has been shown to be crucial to cisplatin resistance in various cancers including non-small-cell lung cancer and ovarian cancer [26,27,28,29]. To determine whether autophagy could be influenced in NB, we measured autophagic vacuole levels in SH-SY5Y cells following treatment with cisplatin, autophagy activator rapamycin, autophagosome inhibitor 3-MA, and autophagosome-lysosome fusion inhibitor HCQ through monodansylcadaverine (MDC) staining and fluorescent microscopy (Figure 1), as well as Cyto-ID staining with image-based cytometry (Figure 2). Immunofluorescent staining for lysosomal-associated membrane protein 1 (LAMP1) and autophagosome marker microtubule-associated proteins 1A/1B light chain 3B (LC3B) following 24-h treatments (Figure 3) allowed for confirmation of results seen with MDC and Cyto-ID. As expected, cisplatin treatment showed autophagy-induction at each of the following time points: 24 h (Figure 3), 48 h (Figure 2), and 72 h (Figure 1). Despite the indication of autophagy upregulation when used alone, rapamycin appeared to have little influence on cisplatin-induced autophagy with levels similar to cisplatin alone (Figure 1, Figure 2 and Figure 3). 3-MA was effective at reducing autophagosome formation both alone and importantly, when combined with cisplatin (Figure 1, Figure 2 and Figure 3) while HCQ increased autophagic vacuole staining, particularly when combined with 0.5 µM cisplatin (Figure 3), likely due to an accumulation of autophagic vacuoles.

### 2.2. Autophagic Induction in U-87 Mg Glioblastoma Cells via TMZ Is Decreased by 3-MA or HCQ Supplementation

Some chemotherapy treatments induce autophagy as a pro-survival mechanism [30]. The objective was to evaluate the effect of known autophagy modulators alongside chemotherapy. We measured autophagic flux through Cyto-ID staining of autophagic vacuoles in U-87 Mg cells treated for 24 h with combinations of TMZ and autophagy regulators (Figure 4). Both TMZ and rapamycin enhanced autophagy compared to control but showed comparable levels to TMZ alone when used in combination. 3-MA was able to inhibit TMZ-induced autophagy while HCQ treatment resulted in increased Cyto-ID staining, likely due to inhibited vesicular fusion. These results were confirmed at 48 h through LAMP1/LC3B immunofluorescence, as shown in Figure 5.

### 2.3. Autophagy Inhibition via 3-MA or HCQ Enhances Chemotherapy-Induced Apoptosis in Neuroblastoma and Glioblastoma

Both autophagy activation and inhibition have been proposed as potential mechanisms for inducing cell death in NB [31,32] and GBM [33,34]. As a result, we sought to examine the effect of autophagy regulators rapamycin, 3-MA, and HCQ in combination with commonly used chemotherapies cisplatin and TMZ using in-vitro models of NB and GBM. This was done through fluorescent staining of SH-SY5Y (Figure 6) and U-87 Mg (Figure 7) for apoptotic markers annexin V (AV) and propidium iodide (PI). In agreement with its lack of autophagic influence in chemotherapy treated cells, rapamycin had little effect on apoptotic levels alone or in combination with cisplatin or TMZ. Treatment with either 3-MA or HCQ alone induced apoptosis compared to control in both NB (Figure 6) and GBM (Figure 7). When combined with cisplatin or TMZ, both autophagy inhibitors enhanced AV/PI staining, with HCQ resulting in considerably increased levels of apoptosis.

### 2.4. Autophagy Inhibition Enhances Reactive Oxygen Species (ROS) Production in SH-SY5Y and U-87 Mg Cells

Due to the enhanced metabolic rates of cancerous cells, elevated levels of ROS are produced, and oxidative damage can be a therapeutic target [35]. As a result, we aimed to analyze the influence of autophagy regulators rapamycin, 3-MA, and HCQ on ROS levels in NB and GBM. We used 2′,7′-dichlorodihydrofluorescein diacetate (H2DCFDA), which diffuses across the cell membrane and is cleaved by intracellular esterases where it can be converted to the fluorescent molecule 2′,7′-dichlorofluorescein (DCF) if oxidized by ROS. This fluorescence was examined via microscopy following 24-h treatments with rapamycin, 3-MA, and HCQ on both NB (Figure 8) and GBM (Figure 9). Autophagy activator rapamycin showed little effect on DCF fluorescence in SH-SY5Y but drastically reduced levels in U-87 Mg cells. Autophagy inhibition via 3-MA or HCQ led to increased ROS production, particularly in SH-SY5Y cells (Figure 8). Results were supported through immunofluorescence with peroxidised lipid marker 4-hydroxynonenal (4-HNE). Minor differences were observed in SH-SY5Y whereby rapamycin reduced, and 3-MA increased fluorescent levels both alone and in combination with cisplatin (Figure 10). In GBM, rapamycin decreased 4-HNE staining when used alone or in combination with TMZ, while 3-MA increased oxidative stress alone but not in combination with TMZ (Figure 11). Furthermore, HCQ had little effect on U-87 Mg alone but drastically enhanced fluorescent levels when combined with 50 µg/mL.

### 2.5. Rapamycin-Induced Autophagy Activation Increases Mitochondrial Stability While Autophagy Inhibition Reduces Mitochondrial Functionality

Mitochondrial vulnerabilities in cancer are another possible target for therapeutic regimens [36]. Our objective was to determine the status of functional mitochondria under various treatments using Tetramethylrhodamine, methyl ester (TMRM), a cell permeant dye that accumulates in active mitochondria with intact membrane potentials. Rapamycin was able to enhance TMRM staining in both cell lines alone and in combination with cisplatin or TMZ. Despite reducing active mitochondria staining alone, 3-MA had little effect in combination with either chemotherapy as did HCQ. However, it should be noted that TMRM staining increased with HCQ treatment alone in SH-SY5Y (Figure 12) while it decreased in U-87 Mg (Figure 13).

### 2.6. Supplementation of Standard Chemotherapies with 3-MA or HCQ Has Little Effect on Apoptotic Levels in Normal Healthy Cells

After demonstrating the anti-neoplastic effects of these autophagy inhibitors, we sought to examine their effect on apoptotic levels in non-cancerous cells. First, Cyto-ID staining for autophagic vacuoles was performed on normal colon mucosal (NCM-460) cells to determine the regulatory capacity of rapamycin, 3-MA, and HCQ at the same doses used in cancer cells (Figure 14). Autophagy inhibitors 3-MA and HCQ had little effect alone while rapamycin and standard chemotherapy TMZ enhanced green fluorescence compared to control. A notable difference was observed when combining TMZ with 3-MA resulting in reduced fluorescent levels comparable to the control. Fluorescent markers AV and PI alongside fluorescent microscopy were used to assess cell death in NCM-460 cells as well as normal human skin fibroblasts (NHF2) in Figure 15 and Figure 16 respectively. Apoptotic levels were slightly raised by 3-MA alone or combined with cisplatin in NCM-460, with little difference observed in combination with TMZ, despite reductions in autophagy levels. Interestingly, despite minor increases in apoptosis when used alone, HCQ reduced AV and PI staining when combined with standard chemotherapeutics. Autophagy overactivation via rapamycin supplementation resulted in slight increases to apoptotic levels when combined with standard chemotherapeutics in NCM-460 cells. The autophagy modulators had little to no effect on apoptosis-induction in NHF2 when used alone or alongside standard chemotherapeutics.

## 3. Discussion

In this report, we have provided new details as to how the activation of autophagy in response to chemotherapy treatment is critical for cancer cell health. Prior research into autophagy modulation alongside standard chemotherapy has provided mixed results with both the activation and inhibition of autophagy proposed as potential anti-cancer treatments. This project aimed specifically to examine and compare the impact of early- and late-stage autophagy inhibition alongside standard chemotherapies using in-vitro models of neuroblastoma and glioblastoma. Furthermore, the roles of oxidative stress and mitochondrial dysfunction in cancer following treatment with chemotherapy and autophagy modulators were examined in these cancers for the first time. By inhibiting autophagic induction with early or late-stage inhibitors, apoptotic levels were selectively enhanced compared to levels normally observed with chemotherapy treatment alone in NB and GBM. Combining 3-MA with standard chemotherapies cisplatin or TMZ inhibited autophagy, destabilized mitochondria, enhanced ROS levels, and induced apoptosis. Late-stage autophagy inhibitor HCQ resulted in the build-up of autophagic vesicles while increasing oxidative stress on the cancer cells, together leading to apoptosis. Though both autophagy inhibitors led to apoptosis in cancer, late-stage autophagy inhibition was more effective at enhancing chemotherapy-induced apoptosis in NB and GBM. Importantly, this work showed for the first time that increased oxidative stress following autophagy inhibition is a potential mechanism through which these autophagy modulators, both early- and late-stage, can induce apoptosis in combination with chemotherapy in NB and GBM. Overall, these autophagic inhibitors are effective at targeting cancer cells through multiple mechanisms in combination with standard chemotherapeutics while having minimal effects on non-cancerous healthy cells.

First off, we confirmed that treatment of SH-SY5Y or U-87 Mg cells with common chemotherapies, cisplatin or TMZ respectively, results in the induction of autophagy [37,38]. Known to be critical for maintenance of cellular homeostasis through protein/organelle recycling, autophagy upregulation is likely serving as a pro-survival mechanism to deal with chemotherapy-associated cell stress [39]. Importantly, we demonstrated the effectiveness of the autophagosome inhibitor 3-MA in reducing autophagy levels in SH-SY5Y following cisplatin treatment. A similar result was observed in U-87 Mg cells when combined with TMZ. Furthermore, HCQ, a known inhibitor of lysosomal fusion, enhanced autophagic vacuole staining due to vesicular build-up. These results indicate that both 3-MA and HCQ could be utilized to reduce chemotherapy-associated autophagic flux in both NB and GBM. On the other hand, although rapamycin-induced autophagy activation was observed, this was rendered virtually ineffective when combined with either cisplatin or TMZ. Figure 17 outlines the process of autophagy and the point of intervention for each of these autophagic modulators.

Due to the complexity and knowledge gaps surrounding autophagy-apoptosis interactions, we sought to analyze the effect of these autophagy regulators in combination with cisplatin or TMZ on apoptosis. Debate continues as to whether reducing a cell’s autophagic defense mechanism or inducing autophagy to cause cell death through lysosomal overactivation or immunogenic cell death is a more effective treatment option [40]. We demonstrated the induction of apoptosis following autophagy inhibition via 3-MA or HCQ treatment alone in both NB and GBM cell lines. As a result, we hypothesized that autophagic inhibition alongside standard chemotherapies could enhance apoptotic cell death. Indeed, a drastic uptick in apoptosis was observed among SH-SY5Y and U-87 Mg cells treated with cisplatin or TMZ respectively when they were combined with 3-MA or HCQ. Late-stage autophagy inhibition via HCQ appeared more effective than early-stage inhibitor 3-MA suggesting its potential use as an anti-cancer supplement. Autophagy upregulation via rapamycin supplementation had little effect on apoptotic levels when used alone or in combination with chemotherapies. As a result, the induction of autophagy seen with cisplatin treatment in NB or TMZ treatment in GBM appears to be serving as a pro-survival mechanism. Fortunately, through 3-MA or HCQ autophagic inhibition, levels of apoptosis are radically increased in SH-SY5Y and U-87 Mg compared to cisplatin or TMZ treatment alone.

To determine whether the cell death enhancement is specific to cancerous cells, we analyzed the effect of the autophagy inhibitors 3-MA and HCQ on two normal cell lines, a healthy skin fibroblast model (NHF2) and healthy colon epithelial cells (NCM-460). Through live-cell fluorescent staining, we observed a slight increase in NCM-460 cell death when combining 3-MA with cisplatin, and a small decrease when combined with TMZ, despite a drastic reduction in autophagic levels. Little to no difference was observed in the NHF2 cells indicating that any change in apoptotic levels in non-cancerous cells is minimal with 3-MA treatment. Interestingly, the large increase in apoptotic levels observed in both cancer lines when adding HCQ to standard chemotherapy was not observed in either healthy cell line when combined with cisplatin or TMZ. This could be a result of reduced potency in normal healthy cells compared to cancerous tissue as a build-up of autophagic vacuoles was not seen with Cyto-ID staining in NCM-460 as it was in both SH-SY5Y and U-87 Mg cell lines. This selectivity of the HCQ-chemotherapy combination to cancerous cells, namely SH-SY5Y and U-87 Mg, compared to healthy cells, NHF2 and NCM-460, indicates that reduced doses of chemotherapy could be used to achieve the same cell death in cancerous tissues, while possibly reducing harmful side-effects.

Although autophagy inhibition itself could be detrimental to cancer cell survival, we aimed to determine whether other common mechanisms of targeting cancer were being exploited by these autophagy inhibitors. Due to the rapidly multiplying nature of cancerous cells and the elevated metabolic rates along with it, cancer cells typically produce elevated amounts of reactive oxygen species (ROS) compared to normal cells. This excessive oxidative stress is critical to both the formation and progression of cancer due to resulting genomic DNA mutations that it produces [41]. However, elevated levels of ROS can be detrimental, resulting in cancer cells becoming particularly susceptible to further oxidative stress due to their already high ROS production [35]. An associated increase in the production of ROS was observed with autophagy inhibition via 3-MA or HCQ, particularly in NB. Conversely, autophagy activation via rapamycin diminished ROS production in GBM while having little effect on NB. Using 4-HNE as a marker of peroxidised lipids, very little difference was observed with rapamycin or HCQ, whether alone or combined with cisplatin. 3-MA resulted in small increases in the number of peroxidised lipids both alone and in combination with cisplatin. In GBM however, rapamycin showed antioxidant capabilities alone and with TMZ, confirming the results from Figure 9. 3-MA resulted in some oxidative stress alone but had very minimal effect when combined with TMZ on U-87 Mg. Interestingly, HCQ, which had very little effect on ROS levels alone in GBM, led to increased oxidative stress when combined with 50 µg/mL TMZ.

Another unique metabolic feature of cancer cells is their preference for lactic fermentation rather than aerobic respiration [36,42]. Dubbed the Warburg Effect, this phenomenon results in an acidified cytoplasm and changes in the MMP, increasing mitochondrial vulnerability. We determined that rapamycin treatment increased the number of active mitochondria in both SH-SY5Y and U-87 Mg cells, likely due to its associated reductions in oxidative stress. When rapamycin induces autophagy, this could result in the clearing of aberrant mitochondria, reducing ROS levels often produced from defective mitochondria. A vicious cycle exists whereby ROS production will further damage other mitochondria which will then produce ROS themselves. As a result, it stands to reason that through autophagy induction, rapamycin can relieve oxidative stress within the cell while increasing the number of viable mitochondria. This trend continued in combination with chemotherapy where the addition of rapamycin was able to enhance TMRM staining and increase mitochondrial health in cancer cells. The inhibition of autophagy through 3-MA or HCQ resulted in reduced mitochondrial health in SH-SY5Y (only 3-MA) and U-87 Mg. However, when combined with chemotherapies cisplatin or TMZ, 3-MA appeared to reduce mitochondrial functionality at lower doses of chemo (0.5 µM cisplatin or 50 µg/mL TMZ) but not at higher doses (2 µM cisplatin or 100 µg/mL TMZ). Alternatively, autophagy inhibition via HCQ appeared to have little effect in combination with any dose of chemotherapy.

Throughout this study, we combined autophagy regulators rapamycin, 3-MA, and HCQ with commonly used chemotherapies cisplatin and TMZ to determine their impact on cellular health and survival. We found that rapamycin induced autophagy in a dose-dependent manner but had little effect when combined with autophagy-inducing chemotherapies. Alone, rapamycin served as an antioxidant while stabilizing mitochondria in both SH-SY5Y and U-87 Mg cells. Its antioxidative benefits were limited when combined with chemotherapy; however, mitochondrial stabilization was still observed. Alternatively, 3-MA reduced autophagy levels while HCQ treatment caused the buildup of autophagic vacuoles both alone and combined with chemotherapy. In addition to autophagy inhibition, 3-MA treatment led to disruptions in MMP while both 3-MA and HCQ induced ROS production. Essentially, autophagy inhibition combined with cisplatin or TMZ treatment in NB and GBM respectively resulted in a noticeable increase in apoptotic levels, particularly with late-stage inhibition. As a result, the combination of autophagy inhibitors with standard chemotherapies should be investigated further as a potential treatment for both neuroblastoma and glioblastoma. These combination treatments could improve patient prognosis/survival as well as quality of life, ameliorating some negative side-effects associated with chemotherapy due to reduced dosing.

## 4. Materials and Methods

### 4.1. Cell Culture

(a)U-87 Mg human glioblastoma cells derived from malignant glioma (ATCC, Cat. No. HTB-14, Manassas, VA, USA) were grown and cultured in Eagle’s Minimum Essential Medium with Earle’s salts and nonessential amino acids (Sigma-Aldrich Canada, Mississauga, ON, Canada) supplemented with 10% (*v*/*v*) fetal bovine serum (Thermo Scientific, Waltham, MA, USA) and 10 mg/mL gentamicin (Gibco BRL, VWR, Mississauga, ON, Canada). Cells were maintained at 37 °C and 5% CO_2_.(b)SH-SY5Y human neuroblastoma cells derived from a metastatic bone tumour (ATCC, Cat. No. CRL-2266, Manassas, VA, USA) were grown and cultured in Dulbecco’s Modified Eagle’s Medium F-12 HAM (Sigma-Aldrich, Mississauga, ON, Canada) supplemented with 10% fetal bovine serum (Thermo Scientific, Waltham, MA, USA) and 10 mg/mL gentamicin (Gibco BRL, VWR, Mississauga, ON, Canada). Cells were maintained at 37 °C and 5% CO_2_.(c)NCM-460 human colon epithelial cells (ATCC, Cat. No. CRL-1831, Manassas, VA, USA) were grown and cultured in Dulbecco’s Modified Eagle Medium (Sigma-Aldrich, Mississauga, ON, Canada) supplemented with 10% fetal bovine serum (Thermo Scientific, Waltham, MA, USA) and 10 mg/mL gentamicin (Gibco BRL, VWR, Mississauga, ON, Canada). Cells were maintained at 37 °C and 5% CO_2_.(d)NHF2 human skin fibroblasts (Coriell Institute for Medical Research, Cat. No. AG09309, Camden, NJ, USA) were grown and cultured in Eagle’s Minimum Essential Medium with Earle’s salts and nonessential amino acids (Sigma-Aldrich Canada, Mississauga, ON, Canada) supplemented with 10% fetal bovine serum (Thermo Scientific, Waltham, MA, USA) and 10 mg/mL gentamicin (Gibco BRL, VWR, Mississauga, ON, Canada). Cells were maintained at 37 °C and 5% CO_2_.

### 4.2. Chemicals and Cell Treatment

(a)Cisplatin (Sigma-Aldrich Canada, Cat. No. PHR1624) was dissolved in 0.9% NaCl to make a 1 mM stock solution. Stock solution was stored at −20 °C until use. SH-SY5Y were treated at doses of 0.5 µM, 1 µM, or 2 µM.(b)Temozolomide (Ontario Chemicals Inc., Cat. No. T1062) was dissolved in dimethylsulfoxide (DMSO) to make a 20 mg/mL stock solution. Stock solution was stored at −20 °C until use. U-87 Mg were treated at doses of 50 µg/mL or 100 µg/mL.(c)Rapamycin (Sigma-Aldrich Canada, Cat. No. R8781) was dissolved in DMSO to make a 40 µM stock. Stock solution was stored at −20 °C until use. SH-SY5Y were treated at doses of 50 nM, 100 nM, or 150 nM. U-87 Mg were treated at doses of 100 nM.(d)3-Methyladenine (Sigma-Aldrich Canada, Cat. No. M9281) was dissolved in milliQ water to make a 50 mM stock. Stock solution was stored at −20 °C until use. SH-SY5Y were treated at doses of 500 µM. U-87 Mg were treated at doses of 200 µM or 500 µM.(e)Hydroxychloroquine (Sigma-Aldrich Canada, Cat. No. H0915) was dissolved in DMSO to make a 3 mg/mL stock. Stock solution was stored at −20 °C until use. SH-SY5Y were treated at doses of 1 µg/mL or 3 µg/mL. U-87 Mg were treated at a dose of 1 µg/mL.

### 4.3. Monodansylcadaverine Staining for Autophagic Vacuoles

MDC, a blue autofluorescent dye, was used to monitor autophagy levels as it accumulates in autophagic vacuoles due to ion trapping and specific lipid membrane interactions. Cells were seeded on 6-well plates 24 h prior to treatment. Following a treatment period of 72 h, cells were incubated for 15 min with 0.1 mM MDC (Sigma-Aldrich Canada, Cat. No. 30432, Mississauga, ON, Canada) dissolved in DMSO. Cells were washed with 1X phosphate buffered saline (PBS) and resuspended in 1X PBS. Cells containing MDC-tagged autophagic vacuoles were detected through epifluorescence microscopy with a Leica DMI6000 B inverted microscope (Leica Microsystems, Concord, ON, Canada). Fluorescence quantification was performed on the images using ImageJ software version 1.52.

### 4.4. Cyto-ID Staining for Autophagic Flux

Cyto-ID selectively labels accumulated autophagic vacuoles monitoring autophagic flux in live cells. Cells were seeded on 6-well plates 24 h prior to treatment. After 24 or 48-h treatments, cells were washed with 1X assay buffer provided in the Cyto-ID autophagy detection kit (Enzo Life Sciences Inc., Cat. No. 51031, Farmingdale, NY, USA), and incubated in half 1:1000 diluted Cyto-ID green detection reagent in Hank’s Balanced Salt Solution and half cell culture media for 30 min at 37 °C. Next, (a) for image-based cytometry, cells were washed and resuspended in 1X assay buffer. A TALI Image-Based Cytometer (Life Technologies Inc., Cat. No. T10796, Burlington, ON, Canada) was utilized to quantify the percentage of cells fluorescing green (containing autophagic vacuoles). Cells from 18 random fields were used to analyze the green (ex. 458 nm; em. 525/20 nm) channel. OR (b) for fluorescent imaging, cells were washed and resuspended in 1X PBS, incubated with 10 µM Hoechst 33342 (Molecular Probes, Cat. No. H3570, Eugene, OR, USA) for 10 min at 37 °C, and imaged using epifluorescence microscopy on a Leica DMI6000 B inverted microscope (Leica Microsystems, Concord, ON, Canada). Fluorescence quantification was performed on the images using ImageJ software version 1.52.

### 4.5. Apoptotic Analysis via Annexin V and Propidium Iodide Staining

AV and PI staining were used to analyze early apoptosis and cell permeabilization indicative of late apoptotic or necrotic cells respectively [43]. Cells were seeded on 8-well slides 24 h prior to treatment. Following 48-h treatments, cells were washed with 1X PBS and resuspended in Annexin V Binding buffer (10 mM HEPES, 140 mM NaCl, 2.5 mM CaCl2, pH 7.4). Annexin V AlexaFluor™ 488 dye (1:20; Life Technologies Inc., Cat. No. A13201, Burlington, ON, Canada), 0.01 mg/mL PI dye (Life Technologies Inc., Cat. No. P3566, Burlington, ON, Canada), and 10 µM Hoechst 3342 (Molecular Probes, Cat. No. H3570, Eugene, OR, USA) were added and cells were incubated in the absence of light for 15 min at 37 °C and 5% CO_2_. Epifluorescence microscopy was used to image the cells using a Leica DMI6000 B inverted microscope (Leica Microsystems, Concord, ON, Canada) and fluorescent quantification was performed with ImageJ software version 1.52.

### 4.6. Measurement of Reactive Oxygen Species

Levels of oxidative stress were measured with H_2_DCFDA (Life Technologies Inc., Cat. No. D-399, Burlington, ON, Canada) which becomes oxidized in the presence of ROS to the fluorescent molecule DCF following acetate group cleavage by intracellular esterases. Cells were seeded in 6-well plates 24 h prior to treatment, and subsequently treated for 24 h. 10 µM H_2_DCFDA dissolved in DMSO was added and cells were incubated in the absence of light at 37 °C and 5% CO_2_ for 30 min. Cells were washed and resuspended in 1X PBS, 10 µM Hoechst 3342 (Molecular Probes, Cat. No. H3570, Eugene, OR, USA) was added, and DCF fluorescence was detected through epifluorescence microscopy with a Leica DMI6000 B inverted microscope (Leica Microsystems, Concord, ON, Canada). ImageJ software was utilized for fluorescent quantification.

### 4.7. Tetramethylrhodamine, Methyl Ester Staining for Healthy Mitochondria

TMRM (Gibco BRL, VWR, Mississauga, ON, Canada) is a cell permeant dye which accumulates in mitochondria with intact membrane potentials, indicative of healthy mitochondria. Cells were seeded on 6-well plates 24 h prior to treatment and subsequently treated for 48 h. Cells were then incubated for 45 min in the absence of light with 100 nM TMRM at 37 °C and 5% CO_2_. Next, cells were washed and resuspended in 1X PBS, followed by the addition of 10 µM Hoechst 3342 (Molecular Probes, Cat. No. H3570, Eugene, OR, USA). Epifluorescence microscopy with a Leica DMI6000 B inverted microscope (Leica Microsystems, Concord, ON, Canada) and ImageJ software were used to visualize and quantify TMRM.

### 4.8. Immunofluorescent Staining

Cells were seeded on 8-chamber slides (Bio Basic Canada Inc., Cat. No. SP41219, Markham, ON, Canada) 24 h prior to treatment. Following treatments of 24 or 48 h, cells were fixed with 3.7% formaldehyde prepared in 1X PBS for 15 min at room temperature, permeabilized with 0.15% Triton X-100 for 2 min and blocked with 5% bovine serum albumin for 1 h. Cells were washed with 1X Tris-Buffered Saline, 0.1% Tween 20 Detergent (TBST) and incubated for 1 h at room temperature with the following primary antibodies: LAMP1 (mouse IgG, 1:250, Cat. No. ab25630) (Abcam Inc., Cambridge, UK), LC3B (rabbit IgG, 1:500, Cat. No. ab192890) (Abcam Inc.), 4-HNE (rabbit IgG, 1:200, Cat. No. ab46545) (Abcam Inc.). Cells were washed with TBST and incubated with goat anti-rabbit Alexa Fluor™ 568 (1:500, Thermo Scientific Canada, Cat. No. A11011) and/or horse anti-mouse fluorescein isothiocyanate (1:500, MJS BioLynx Inc., Brockville, Canada, Cat. No. Fl-2000) secondary antibodies for 1 h at room temperature. Cells were washed with TBST and incubated for 2 min with 10 µM Hoechst 3342 (Molecular Probes, Cat. No. H3570, Eugene, OR, USA). Cells were washed with 1X PBS and imaged via epifluorescence microscopy with a Leica DMI6000 B inverted microscope (Leica Microsystems, Concord, ON, Canada). ImageJ software was utilized for fluorescent quantification.

### 4.9. Statistical Testing

Statistical testing was conducted with GraphPad Prism 6 statistical software with p-values less than 0.05 taken as significant. One-way analysis of variance was used for comparing group means to control or other groups.

## Figures and Tables

**Figure 1 ijms-24-12052-f001:**
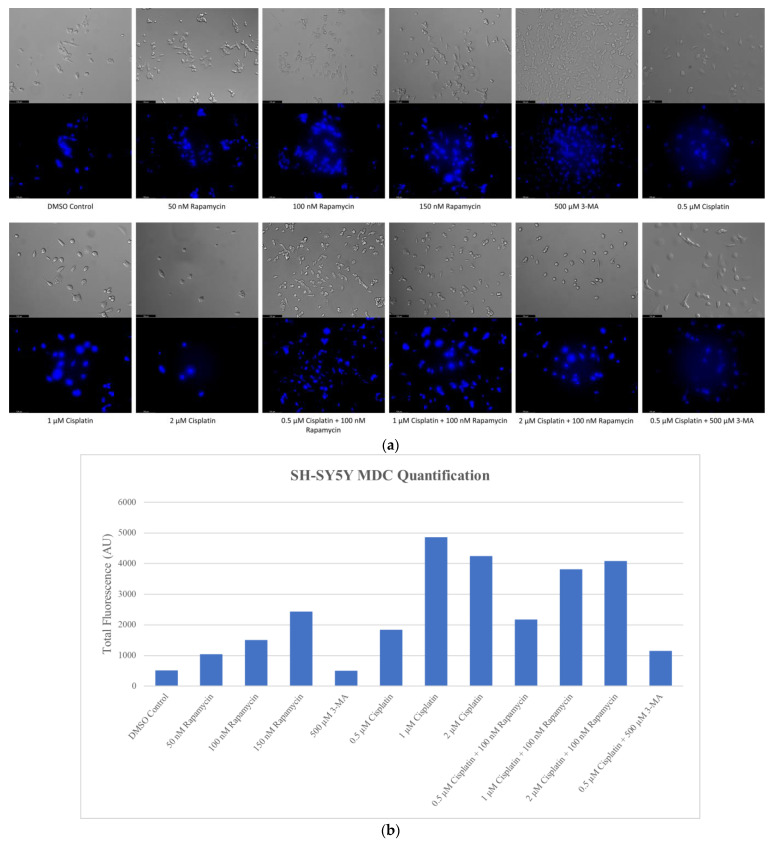
Cisplatin induces autophagy in a dose-dependent manner in SH-SY5Y cells. (**a**) Live cell staining of SH-SY5Y for MDC (blue) and (**b**) fluorescent quantification following 72-h individual and combination treatments with chemotherapy cisplatin, autophagy activator rapamycin, and autophagy inhibitor 3-MA. Cisplatin or rapamycin treatment resulted in increased MDC staining in a dose-dependent manner indicative of enhanced autophagy. Autophagy levels in combined cisplatin and rapamycin treatments were comparable to cisplatin alone. 3-MA reduced cisplatin-induced autophagy despite little observable effect with 3-MA treatment alone. This experiment is designed to show qualitative manipulation of autophagy with fluorescence measurements using ImageJ software version 1.52. Micrographs were taken at 200× magnification. Scale bar = 100 µm.

**Figure 2 ijms-24-12052-f002:**
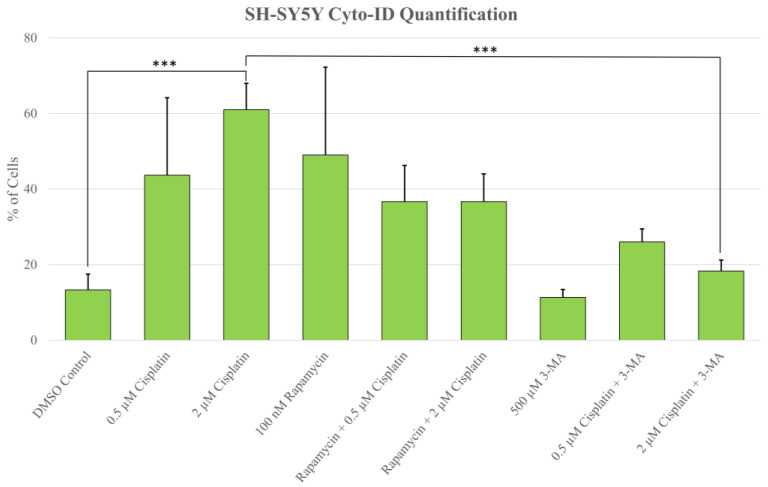
3-MA significantly reduces cisplatin-induced autophagy in SH-SY5Y. Live cell fluorescent staining quantification of SH-SY5Y for Cyto-ID autophagy indicator was conducted following 48-h treatments with cisplatin, rapamycin, and 3-MA. Cisplatin induced autophagy compared to control while combinations with rapamycin were comparable to cisplatin alone. 3-MA in combination with cisplatin reduced autophagy levels compared to cisplatin alone. Results were obtained using image-based cytometry to assess the percentage of cells with active autophagy compared to control. Values are expressed as mean ± SD from 3 independent experiments. Statistical calculations were performed using single factor analysis of variance. *** *p* < 0.001.

**Figure 3 ijms-24-12052-f003:**
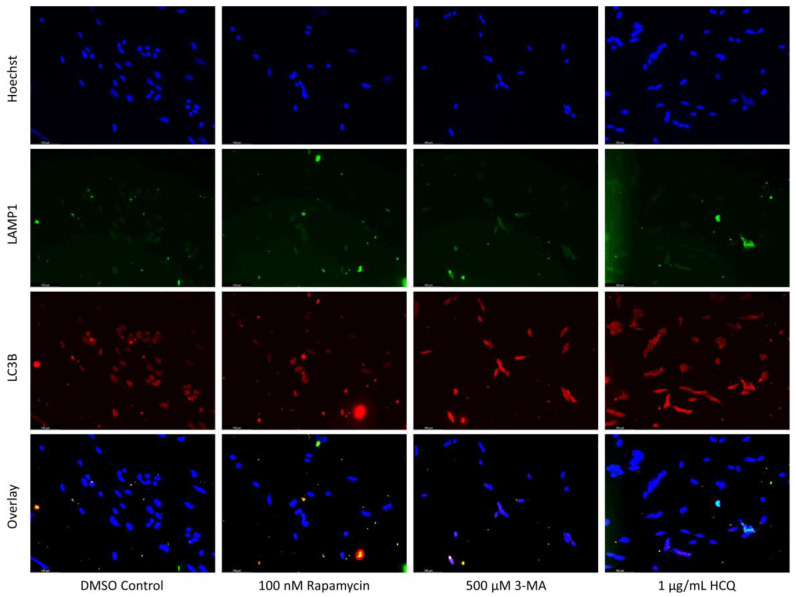
HCQ causes autophagic vacuole accumulation in SH-SY5Y. (**a**) Immunofluorescent staining of SH-SY5Y for lysosomal marker LAMP1 (green) and autophagosome marker LC3B (red) and (**b**) fluorescent quantification following 24-h individual and combination treatments with cisplatin, rapamycin, 3-MA, and HCQ. Rapamycin enhanced lysosomal formation while HCQ led to a build-up of autophagosomes when used alone. 0.5 µM cisplatin upregulated both autophagy markers while 3-MA supplementation reduced LAMP1 and LC3B staining. HCQ had little effect on autophagy levels following cisplatin treatment. Nuclei were counterstained with Hoechst (blue). This experiment is designed to show qualitative manipulation of autophagy with fluorescence measurements using ImageJ software version 1.52. Micrographs were taken at 200× magnification. Scale bar = 100 µm.

**Figure 4 ijms-24-12052-f004:**
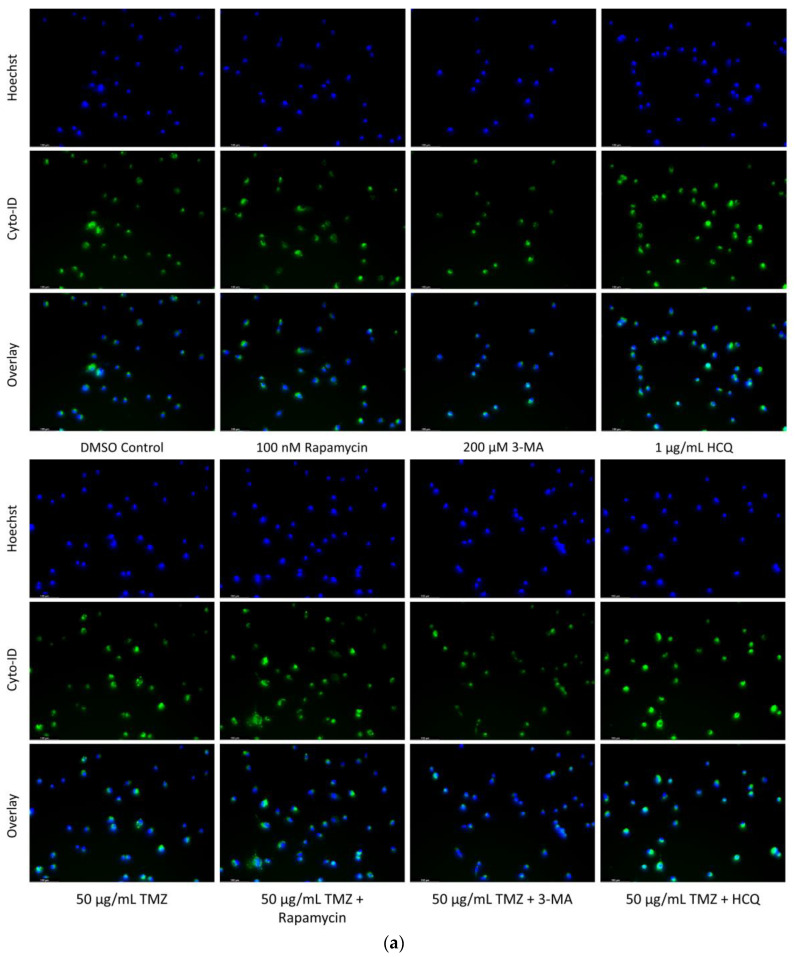
Rapamycin induces autophagy in U-87 Mg alone but has limited effect when combined with TMZ. (**a**) Cyto-ID (green) fluorescent staining of U-87 Mg and (**b**) fluorescent quantification following 24-h treatments with a standard chemotherapy TMZ, autophagy activator rapamycin, and autophagy inhibitors 3-MA and HCQ. TMZ, rapamycin, and HCQ alone led to increased fluorescent staining while 3-MA had little effect compared to the DMSO control. In combination with TMZ, rapamycin had little effect on autophagy levels while 3-MA reduced and HCQ increased Cyto-ID fluorescence. Nuclei were counterstained with Hoechst (blue). This experiment is designed to show qualitative manipulation of autophagy with fluorescence measurements using ImageJ software version 1.52. Micrographs were taken at 200× magnification. Scale bar = 100 µm.

**Figure 5 ijms-24-12052-f005:**
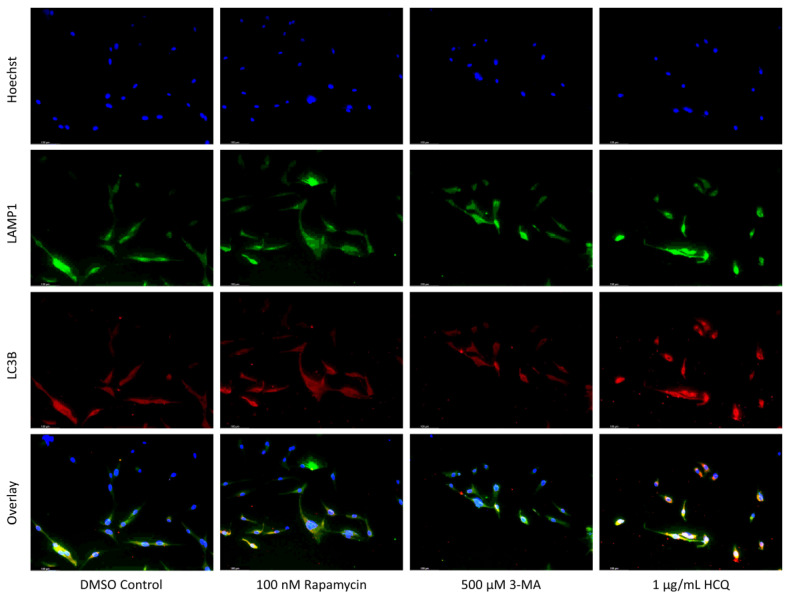
HCQ treatment in U-87 Mg leads to the build-up of autophagic vacuoles. (**a**) Immunofluorescent staining of U-87 Mg for LAMP1 (green) and LC3B (red) and (**b**) fluorescent quantification following 48-h treatments with TMZ, rapamycin, 3-MA, and HCQ. Rapamycin and 50 µg/mL TMZ treatments led to increased LC3B levels indicative of upregulated autophagy. 3-MA treatment slightly reduced autophagic levels while HCQ led to the build-up of autophagosomes. In combination with 50 µg/mL TMZ, rapamycin and 3-MA had minor influence while HCQ led to drastic autophagic vacuole build-up. 100 µg/mL TMZ did not induce autophagy but LC3B and LAMP1 fluorescence increased when rapamycin or HCQ was supplemented as well. Nuclei were counterstained with Hoechst (blue). This experiment is designed to show qualitative manipulation of autophagy with fluorescence measurements using ImageJ software version 1.52. Micrographs were taken at 200× magnification. Scale bar = 100 µm.

**Figure 6 ijms-24-12052-f006:**
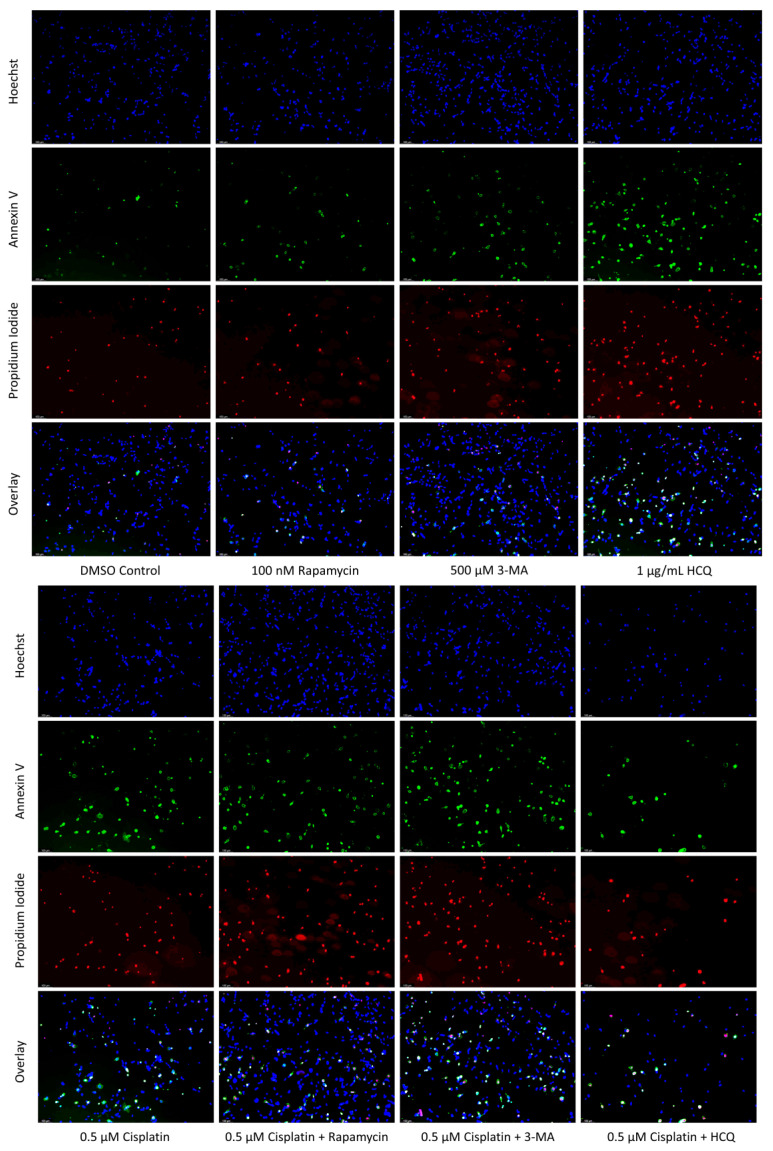
Autophagy inhibition via 3-MA or HCQ induces apoptosis alone or combined with cisplatin in SH-SY5Y. (**a**) AV (green) and PI (red) fluorescent staining and (**b**) fluorescent quantification of SH-SY5Y cells treated with cisplatin, rapamycin, 3-MA, and HCQ for 48 h was conducted to determine apoptotic levels. 3-MA and HCQ alone or combined with cisplatin led to increased apoptosis compared to control or cisplatin alone. Rapamycin had little effect on apoptotic levels alone with slightly increased levels when combined with 0.5 µM cisplatin. Nuclei were counterstained with Hoechst (blue). This experiment is designed to show qualitative apoptosis levels with fluorescence measurements using ImageJ software version 1.52. Micrographs were taken at 100× magnification. Scale bar = 100 µm.

**Figure 7 ijms-24-12052-f007:**
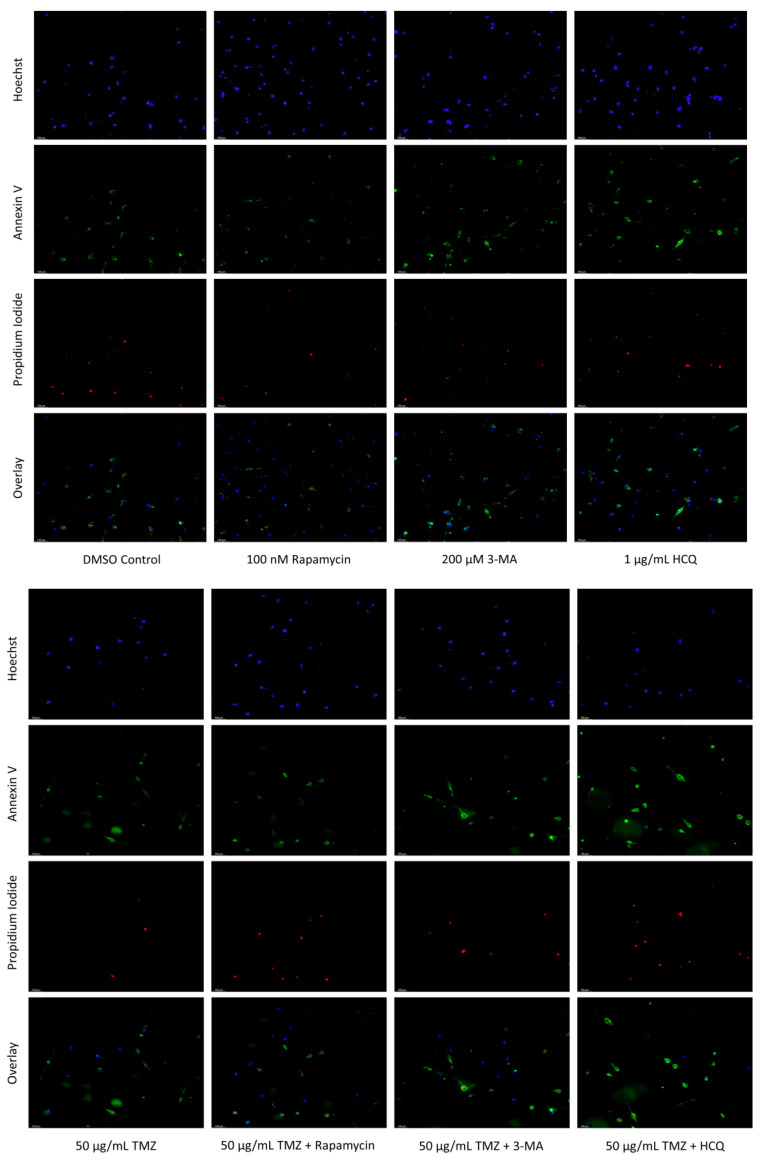
3-MA and HCQ enhance TMZ induced apoptosis in-vitro. (**a**) Annexin V (green) and propidium iodide (red) fluorescent staining and (**b**) fluorescent quantification of U-87 Mg cells treated with TMZ, rapamycin, 3-MA, and HCQ for 48 h to examine apoptosis. Autophagy inhibitors 3-MA and HCQ induced apoptosis alone while rapamycin had little effect. In combination with TMZ, both 3-MA and HCQ further induced apoptosis. Nuclei were counterstained with Hoechst (blue). This experiment is designed to show qualitative apoptosis levels with fluorescence measurements using ImageJ software version 1.52. Micrographs were taken at 100× magnification. Scale bar = 100 µm.

**Figure 8 ijms-24-12052-f008:**
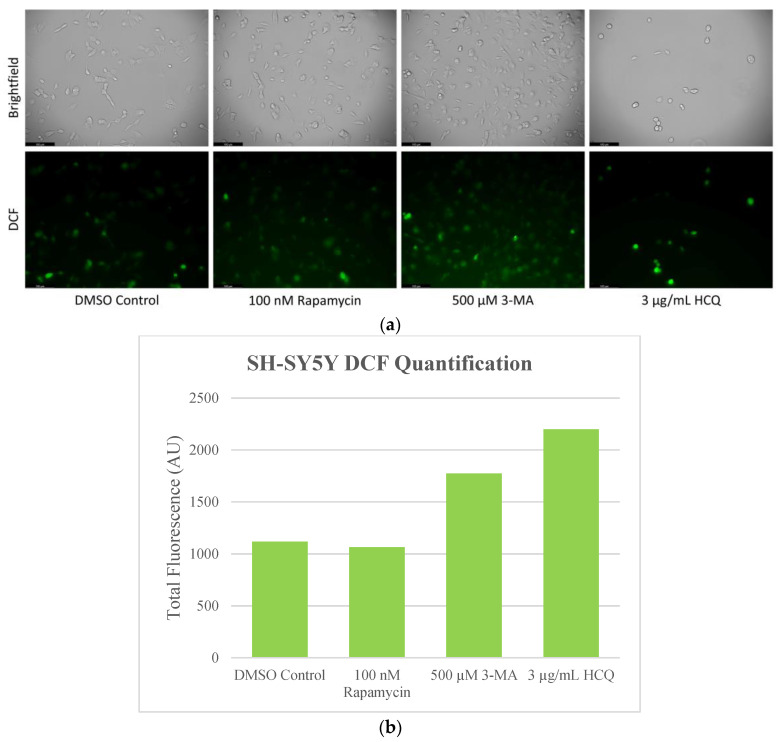
Autophagy inhibition induces oxidative stress in SH-SY5Y cells. (**a**) Live cell staining of SH-SY5Y for oxidative stress marker DCF (green) and (**b**) fluorescent quantification following 24-h treatments with autophagy regulators rapamycin, 3-MA, and HCQ was performed. Rapamycin had no effect on DCF staining while autophagy inhibition by 3-MA or HCQ increased levels of ROS. This experiment is designed to show qualitative ROS levels with fluorescence measurements using ImageJ software version 1.52. Micrographs were taken at 200× magnification. Scale bar = 100 µm.

**Figure 9 ijms-24-12052-f009:**
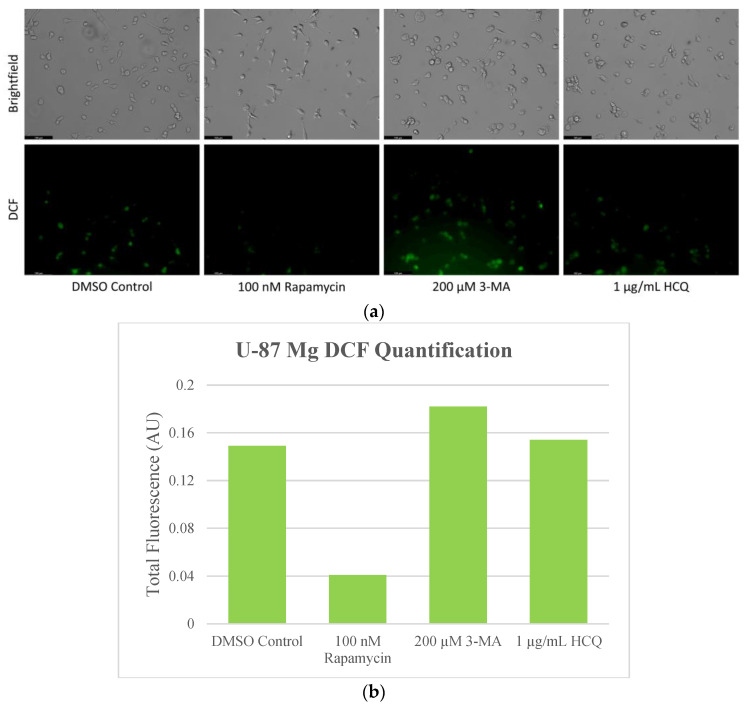
Rapamycin-induced autophagy decreases basal ROS levels in U-87 Mg cells. (**a**) Live cell staining of U-87 Mg for ROS marker DCF (green) and (**b**) fluorescent quantification following 24-h treatments with autophagy regulators rapamycin, 3-MA, and HCQ was performed. Rapamycin drastically reduced DCF fluorescence while both 3-MA or HCQ slightly enhanced oxidative stress levels. This experiment is designed to show qualitative ROS levels with fluorescence measurements using ImageJ software version 1.52. Micrographs were taken at 200× magnification. Scale bar = 100 µm.

**Figure 10 ijms-24-12052-f010:**
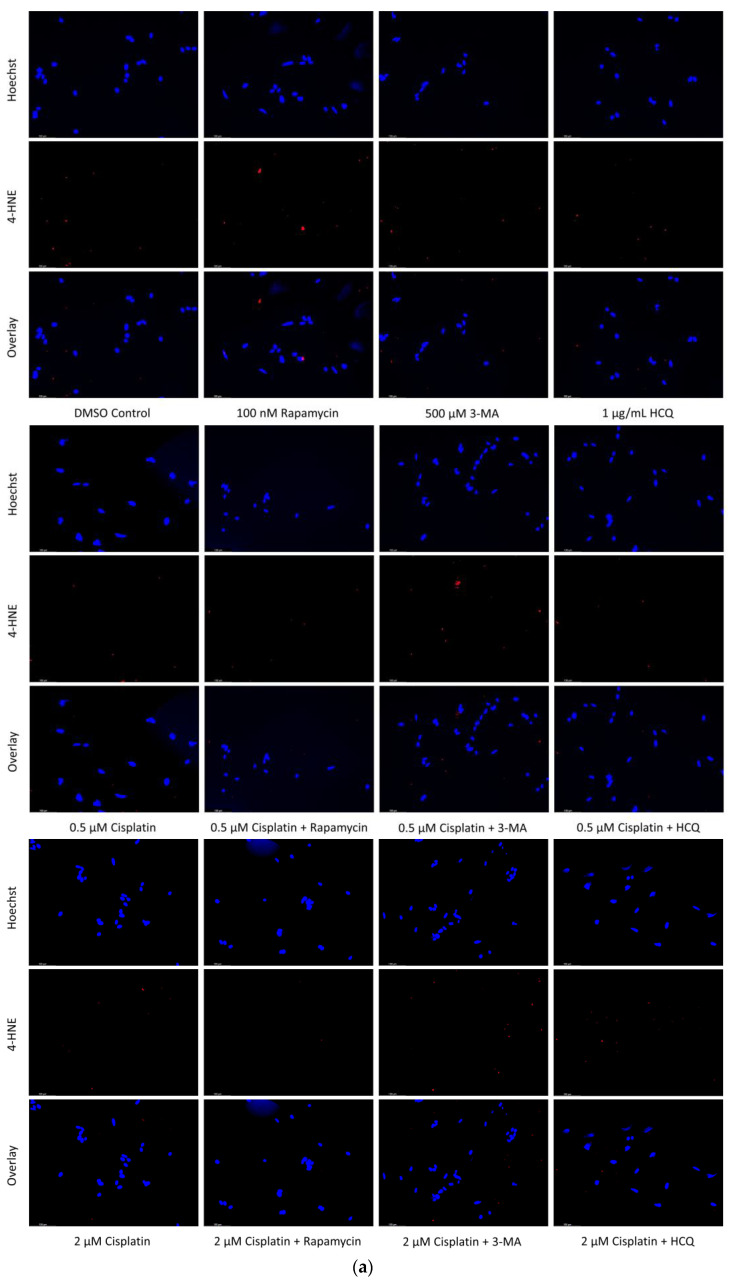
Autophagy modulation has little influence on lipid peroxidation marker 4-HNE staining in SH-SY5Y. (**a**) Immunofluorescent staining of SH-SY5Y for 4-HNE (red) and (**b**) fluorescent quantification following 24-h treatments with cisplatin, rapamycin, 3-MA, and HCQ was performed. Autophagy inhibition via 3-MA supplementation led to slight increases in 4-HNE staining while rapamycin slightly reduced fluorescent levels both alone and in combination. Nuclei were counterstained with Hoechst (blue). This experiment is designed to qualitatively show levels of oxidative stress with fluorescence measurements using ImageJ software version 1.52. Micrographs were taken at 200× magnification. Scale bar = 100 µm.

**Figure 11 ijms-24-12052-f011:**
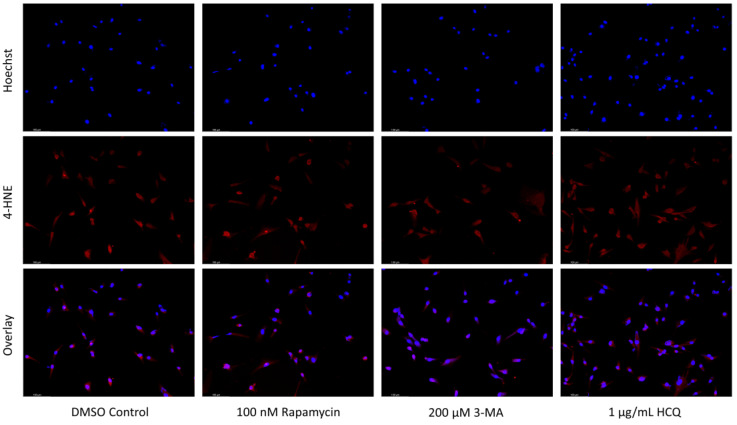
Autophagy upregulation reduces lipid peroxidation marker 4-HNE while inhibition enhances oxidative stress levels in GBM. (**a**) Immunofluorescent staining of U-87 Mg for 4-HNE (red) and (**b**) fluorescent quantification following 24-h treatments with TMZ, rapamycin, 3-MA, and HCQ were conducted. Rapamycin treatment was able to reduce fluorescent levels and oxidative stress when used alone or in combination with TMZ. 3-MA increased 4-HNE staining alone but had little effect on ROS production when combined with TMZ. HCQ had no effect on oxidative stress when used alone but drastically increased staining when combined with 50 µg/mL TMZ. Nuclei were counterstained with Hoechst (blue). This experiment is designed to qualitatively show levels of oxidative stress with fluorescence measurements using ImageJ software version 1.52. Micrographs were taken at 200× magnification. Scale bar = 100 µm.

**Figure 12 ijms-24-12052-f012:**
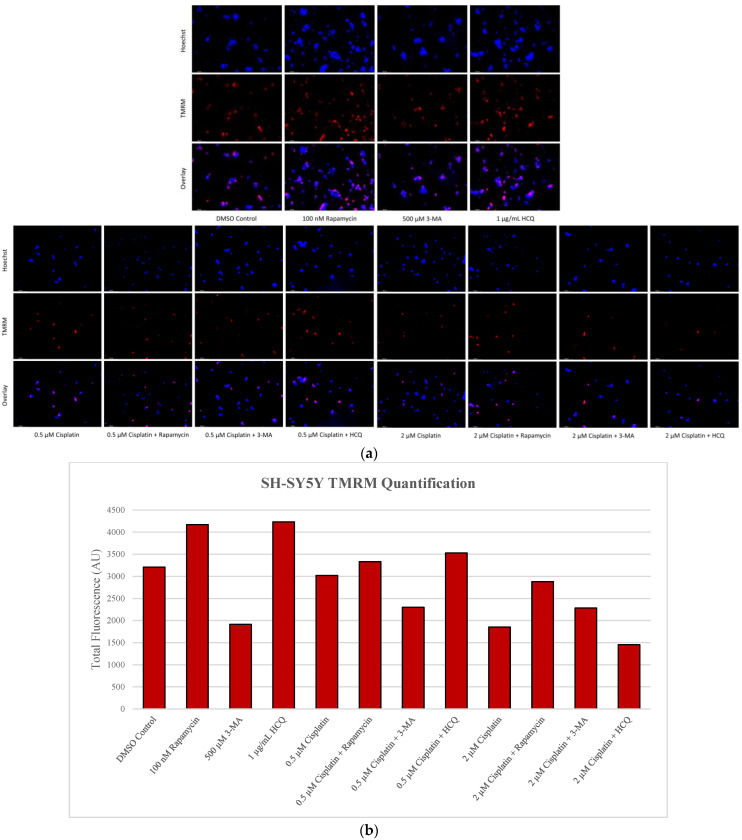
Early-stage autophagy inhibition by 3-MA diminishes MMP in SH-SY5Y. (**a**) Live cell staining of SH-SY5Y for mitochondrial marker TMRM (red) and (**b**) fluorescent quantification following 48-h treatments with cisplatin, rapamycin, 3-MA, and HCQ was conducted. Rapamycin and HCQ treatment enhanced TMRM fluorescence while 3-MA treatment drastically reduced TMRM staining. 0.5 µM cisplatin had little effect on mitochondrial stability but showed reduced staining when combined with 3-MA. Rapamycin increased TMRM fluorescence when combined with cisplatin. Nuclei were counterstained with Hoechst (blue). This experiment is designed to qualitatively show changes in the mitochondrial membrane potential with fluorescence measurements using ImageJ software version 1.52. Micrographs were taken at 200× magnification. Scale bar = 100 µm.

**Figure 13 ijms-24-12052-f013:**
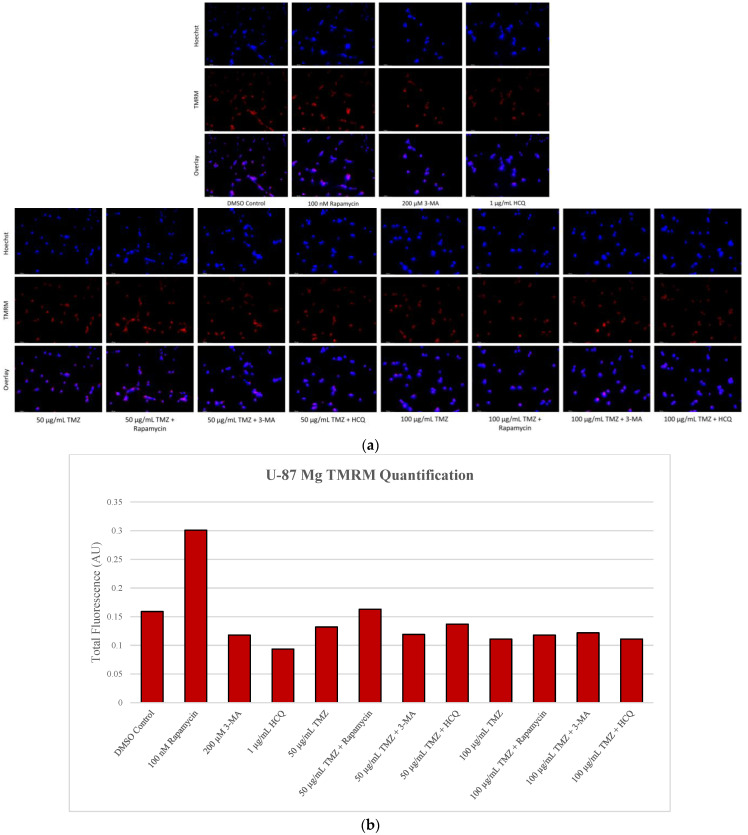
Autophagy activation enhances TMRM fluorescence in U-87 Mg. (**a**) TMRM (red) staining of U-87 Mg and (**b**) fluorescent quantification following 48-h treatments with TMZ, rapamycin, 3-MA, and HCQ was performed. Rapamycin treatment increased TMRM staining both alone and in combination with TMZ. Both 3-MA and HCQ reduced fluorescent levels alone but had little effect when combined with TMZ. Nuclei were counterstained with Hoechst (blue). This experiment is designed to qualitatively show changes in the mitochondrial membrane potential with fluorescence measurements using ImageJ software version 1.52. Micrographs were taken at 200× magnification. Scale bar = 100 µm.

**Figure 14 ijms-24-12052-f014:**
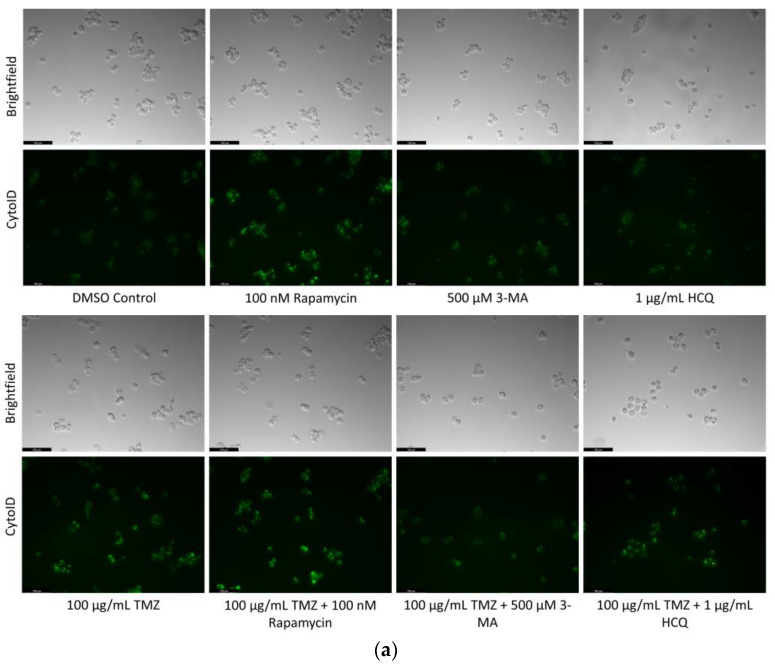
Autophagy inhibitors 3-MA and HCQ have little influence on autophagic flux in non-cancerous cells. (**a**) Cyto-ID (green) staining of NCM-460 cells and (**b**) fluorescent quantification after 48-h treatment with combinations of TMZ, rapamycin, 3-MA, and HCQ was done. Rapamycin alone induced notable amounts of autophagy as indicated by the enhanced green fluorescence compared to the control, while 3-MA and HCQ had little effect. TMZ alone and combined with rapamycin or HCQ induced large amounts of autophagy compared to control while 3-MA reduced levels of green fluorescence comparable to the control. This experiment is designed to show qualitative manipulation of autophagy with fluorescence measurements using ImageJ software version 1.52. Micrographs were taken at 200× magnification. Scale bar = 100 µm.

**Figure 15 ijms-24-12052-f015:**
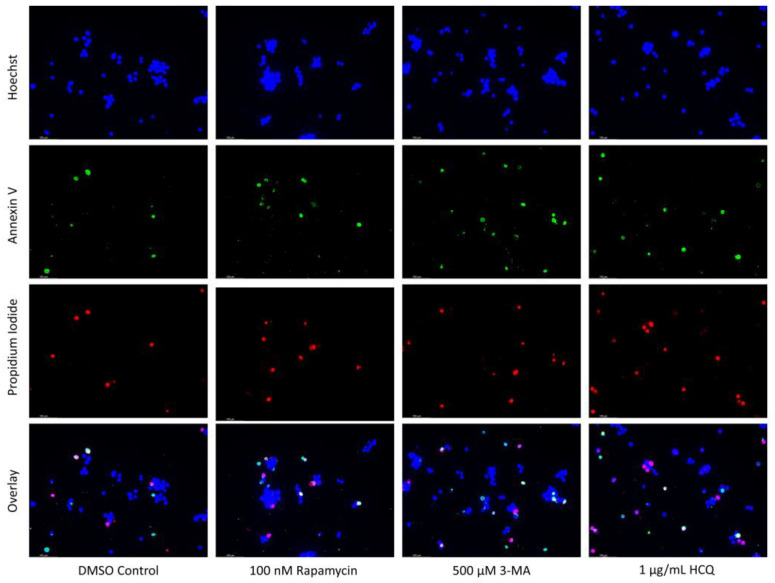
Little apoptosis induction occurs in NCM-460 following 3-MA or HCQ treatment. (**a**) Live cell staining for AV (green) and PI (red) and (**b**) fluorescent quantification in normal colon mucosal cells (NCM-460) was conducted subsequent a 48-h treatment with chemotherapies cisplatin and TMZ alone and combined with rapamycin, 3-MA, and HCQ. 3-MA slightly increased the total fluorescence when used alone or combined with cisplatin, while having little effect combined with TMZ. HCQ enhanced fluorescent staining alone but reduced the amount of staining when combined with either cisplatin or TMZ. This experiment is designed to show qualitative apoptosis levels with fluorescence measurements using ImageJ software version 1.52. Micrographs were taken at 200× magnification. Scale bar = 100 µm.

**Figure 16 ijms-24-12052-f016:**
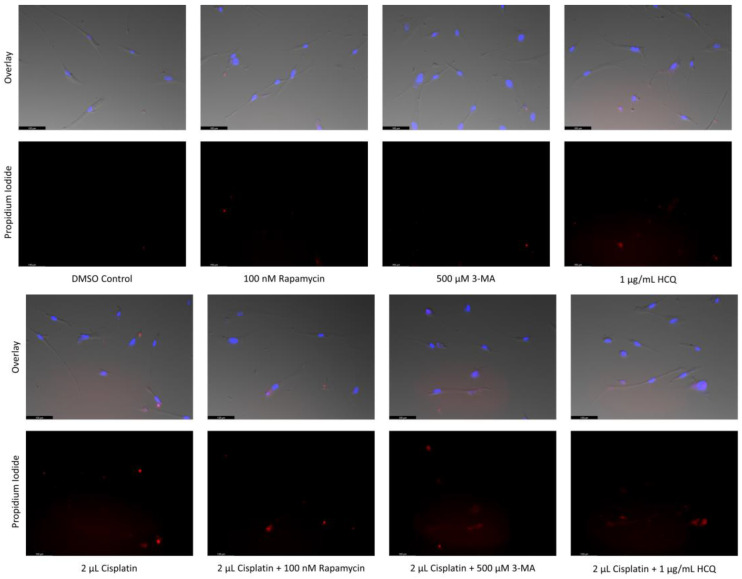
3-MA and HCQ have little influence on cell morphology and apoptosis in normal skin fibroblasts. Propidium iodide staining of human skin fibroblast cells (NHF2) following treatment for 48 h with cisplatin, TMZ, and autophagy regulators rapamycin, 3-MA, and HCQ was performed. Little difference was observed in fluorescent levels or morphology when autophagy regulators were added alone compared to control, or with either chemotherapy compared to the chemotherapy alone. This experiment is designed to show qualitative apoptosis levels with fluorescence measurements using ImageJ software version 1.52. Micrographs were taken at 200× magnification. Scale bar = 100 µm.

**Figure 17 ijms-24-12052-f017:**
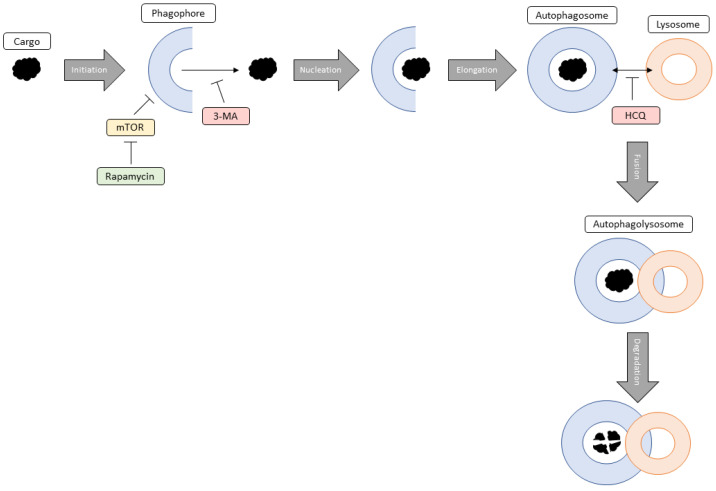
General diagram outlining the steps of autophagy and the modulation by rapamycin, 3-MA, and HCQ. Initiation of autophagy involves the formation of the double-membraned phagophore in response to a cargo that needs degrading. Rapamycin targets this step through the inhibition of mTOR resulting in phagophore formation. Nucleation occurs when autophagic proteins localize to the growing phagophore. 3-MA can inhibit this process by blocking class III phosphatidyl 3-kinase, normally involved in vesicle nucleation. During elongation, the phagophore grows into an autophagosome which traps the cargo inside. Fusion involves the joining of the autophagosome with an acidic lysosome forming an autophagolysosome. The late-stage autophagy inhibitor HCQ prevents this fusion from occurring resulting in the build-up of these autophagic vacuoles. Finally, degradation results in the lysosomal breakdown of the desired cargo because of low pH.

## Data Availability

The data presented in this study are available.

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
