# Peer review of "Autophagy Inhibition via Hydroxychloroquine or 3-Methyladenine Enhances Chemotherapy-Induced Apoptosis in Neuro-Blastoma and Glioblastoma"

_ijms, 2023, doi:10.3390/ijms241512052_

Round 1

Reviewer 1 Report

Neuroblastoma is the most common tumour in children under 1 year old, accounting for 13 approximately 50% of infant cancer cases. 

I am not sure if this statistic is correct. Please check this.

The authors propose combining chemotherapy in NB and GMB with autophagy modulators, for example, autophagy. The following combination has been proposed:

combining hydroxychloroquine with 0.5 μ M cisplatin or 50 μ g/mL temozolomide was 25 as or more effective than 2 μ M cisplatin or 100 μ g/mL temozolomide alone.

Treatment options have been covered in lines 48-53 for both types of cancers.

A good narration has been provided for the process of autophagy (mainly the cellular process but not so much the key molecular players)

I wouldn’t expect to see the paper aims in the middle of the introduction section but rather at the very end of this section. This section though is not the aim so the authors could omit this sentence and just carry on with introducing the 3 autophagy modulators.

The role of rapamycin 3-MA and HCQ have also been introduced.

In the paragraph starting with line 89, the authors could make clear the early and later cancer-stage effects of autophagy. Some previous works have been introduced that combine cisplatin/ TMZ with HCQ.

We hypothesize that the inhibition of autophagy alongside standard chemotherapy regimens will result in enhanced apoptotic levels and could be utilized to reduce chemotherapy doses in NB and adverse effects 118 in children, as well as increase the lifespan/quality of life for GBM patients. 

It is not clear to me what this study is testing that has not been tested before. Autophagy will get activated post-chemotherapy to desensitise the cells to this agent while inhibiting autophagy will increase the sensitivity of these cells to the agent. What is novel then?

The study is devoid of any statistical analysis, or any description of what statistical tests have been performed.

Figure 1-3: The authors attempt to quantify fluorescence pertaining to MDC, cytoID, LAMP1 and LC3B following a combination of drug treatments.  As things stand, none of the quantification made is of any scientific value since no statistical tests were performed at all. Please rework these figures, perform statistics and analyse the results. Figure 2 does seem to have statistical work but kindly expand on what test has been performed and explain this in the methods and results. If the results stand the test of statistical analyses, then this is a good result (rapamycin not doing much in combination but 3-MA/HCQ reduce autophagy in combination with drugs).

Figure 4: Again, showing cyto-ID as a readout for the test. With the same results as above but in GBM. Please perform a statistical analysis, otherwise, none of the results obtained is of any scientific value.

The readouts used in this manuscript are good but the authors need to explain what each readout signifies for example LAMP1/LC3B was not explained.  Figure 5 needs more explanation and also there doesn’t seem to be a straightforward trend with the HCQ function.

Please perform statistical analyses for all figures. None of the conclusions made are valid until they are statistically tested.

Figure 6-7 testing apoptosis levels in the same drug treatment, again showing the known premise that inhibiting autophagy increases apoptosis and enhances the sensitivity of cells to drug treatment in NB and GBM. Please perform stats.

Figures 8-9, The effect of autophagy inhibition on reducing ROS in NB and GBM. The same comment as before, please add stats. If the results stand the statistical test, then they are interesting and useful.

In figures 10-11, again there is no valid statement made without statistical testing, one can’t just look at a table or figure and assume differences!

In line 324 the narrative changes to my objective, as if this data was taken from a thesis or the like, please kindly make sure your narrative and voice are consistent.

12-13: Again, for this, no stats were done. TMRM is a readout for healthy mitochondria but despite not having a statistical test, there does not seem to be a trend with these drugs. The authors suggest a mixed response for HCQ in NB and GBM, what does this signify?

14-15: The control cell line and the data produced need statistical testing please what is the purpose of this (effect on non-malignant cells)? Don’t see side effects happen completely unexpectedly and not be related to autophagy? Please perform statistical testing. 

My question again is what aspects of this study are novel and add to the scientific literature?

Some minor issues here and there.

Author Response

Thank you for your review of our MS and for providing valuable comments/suggestion. I have attached a MS work file with response to your comments. 

Reviewer 2 Report

The subject of the article about standard chemotherapeutic agents currently used in routine clinical treatment is quite interesting. The authors demonstrated in-vitro further induction of apoptosis by inhibition of autophagy by the administration of reduced doses of chemotherapeutics, cisplatin and temozolomide, in the treatment of neuroblastoma and glioblastoma. It will be possible to reveal whether the results of the study may be clinically useful after in-vivo verification. It is recommended to arrange the figures by showing them as standard deviation or standard error in the graphic results.

Author Response

Thank you for your encouraging comments and valuable suggestions. I have attached our response to your comments.

Round 2

Reviewer 1 Report

The authors have addressed my comments, but one issue still remains.

The authors state that many of their plots constitute qualitative data (for example total fluorescent measurement). The authors have quantified fluorescence intensity (AU: arbitrary unit) levels but could then further process this by providing plots for relative fluorescence levels (test/ control) and on these they can perform statistics.

This paper below has performed cyto-ID (figure 2) and then they calculate relative CytoID by dividing test/ control to measure autophagic compartment size and on that they perform statistics (figure 3).

https://pubmed.ncbi.nlm.nih.gov/25714620/

By this token, the AU readings in your figures should be convertible to a fold change by dividing by control and this should then be easily comparable. 

minor proof-reading

Author Response

Thank you for your insight and for providing the reference. We have gone through these figures you suggested in the reference, and it appears the authors have measured the fluorescent intensity at different locations on the same image, then converted it to relative fluorescence compared to the control for statistics. In our figures, we have quantified the fluorescence on the figure as a whole, rather than choosing to select multiple regions within one image. We feel it would not add anything to take multiple readings from the same image as measuring the same sample repeatedly and comparing to the same control does not show anything different than what we are already showing qualitatively. One could take multiple images at different times and take the intensities to obtain a standard error and statistics, but with our 15 figures, some of which have multiple stains, this amount of work is not possible in a minor revision. Rather, our intention was to show the very obvious qualitative difference through multiple confirmatory stains.